# InteractComp: Evaluating Search Agents With Ambiguous Queries

## Abstract

Language agents are increasingly deployed for web search, yet most benchmarks assume queries are fully specified and unambiguous. In practice, user queries are often incomplete and require clarification before accurate answers can be produced. To systematically evaluate this overlooked capability, we introduce Inter-actComp, a benchmark explicitly designed to evaluate whether agents can recognize and resolve such ambiguity by deciding when to search, when to ask clarifying questions, and when to answer. InteractComp contains 210 expert-curated questions spanning 9 domains, constructed through a systematic target-distractor methodology that ensures genuine ambiguity and controlled disambiguation. Extensive experiments on 17 models reveal striking behavioral patterns: even state-of-the-art models achieve less than 14% accuracy, not because they lack reasoning ability, but because they exhibit systematic overconfidence and underutilize interaction opportunities. Ablation and forced-interaction analyses confirm this bottleneck: when compelled to interact, models achieve significant performance gains, demonstrating latent capacity that current strategies fail to unlock. A longitudinal study further highlights a blind spot in model development, while retrieval benchmarks show rapid improvement, interactive capabilities remain stagnant. By exposing this overlooked weakness, InteractComp provides not only a diagnostic tool but also a foundation for designing agents that are uncertainty-aware, strategically interactive, and aligned with real-world user behavior.

## 1 Introduction

Language agents have demonstrated remarkable potential for information retrieval in the digital world (Mialon et al., 2023; Chen et al., 2025a; Wei et al., 2025; Zheng et al., 2025). Search agents (Zheng et al., 2025; Li et al., 2025; Wu et al., 2025) can handle complex user queries by actively decomposing them and performing search and browse actions across the internet to gather information. However, they face a fundamental challenge in real-world deployment: human search behavior is typically iterative rather than comprehensive. Users often begin with ambiguous queries and progressively refine them through interaction, yet current benchmarks assume complete query specification from the outset.

This mismatch poses significant obstacles for practical deployment, as agents that cannot handle ambiguous queries will make incorrect assumptions about user intent, pursuing irrelevant search paths and wasting resources. Existing benchmarks fall into two categories with distinct limitations: interaction benchmarks (Qian et al., 2024; Yao et al., 2024; Luo et al., 2025) focus primarily on general conversational settings rather than goal-oriented search tasks, while search benchmarks (Mialon et al., 2023; Wei et al., 2025; Zhou et al., 2025) excel at complex reasoning but consistently assume users can articulate their information needs precisely. This creates a significant evaluation gap: current benchmarks cannot assess agents' ability to handle the common scenario where users begin with incomplete information needs and must iteratively refine their queries through strategic collaboration with the agent.

Constructing benchmarks for ambiguous query handling presents significant challenges, as queries must appear reasonable yet lack sufficient information for accurate resolution. User ambiguity is particularly pronounced when dealing with similar concepts that share overlapping attributes. Inspired by this observation, we design a construction strategy that systematically pairs an obscure

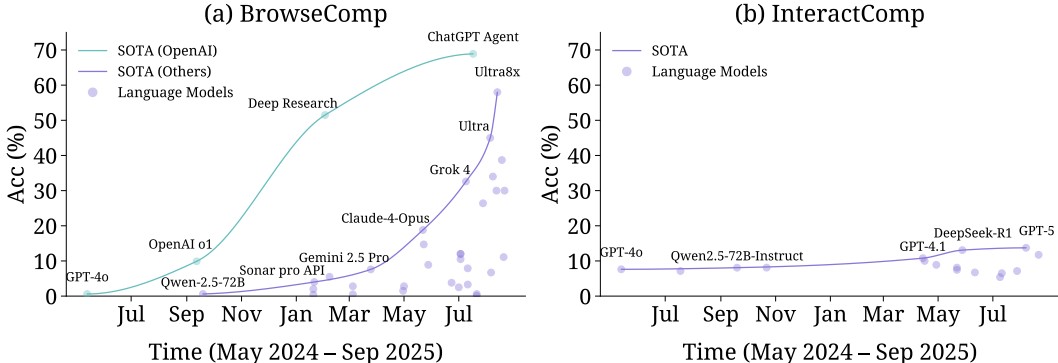

Figure 1: Evolution of different search agent capabilities over time. (a) BrowseComp demonstrates rapid progress on well-specified queries requiring no interaction. (b) INTERACTCOMP reveals stagnation in handling ambiguous queries requiring strategic interaction.

target entity with a similar popular entity, crafting questions using only their shared attributes while withholding distinctive information. This methodology reveals systematic overconfidence in language models by creating scenarios where direct answering fails and strategic interaction becomes necessary.

To address this evaluation gap, we introduce INTERACTCOMP, a benchmark specifically designed to test search agents' ability to handle ambiguous queries through strategic interaction. INTERACTCOMP contains 210 expert-curated questions spanning 9 domains, each following this construction paradigm. Each dataset instance contains an ambiguous question, contextual information for disambiguation, the correct answer, along with domain and identifier metadata.

We evaluate 17 models using the ReAct framework across three action spaces: direct answering, search-augmented responses, and full interaction capabilities. Results reveal profound limitations: even GPT-5 achieves only 13.73% accuracy, while most models struggle to reach double-digit performance. The results expose systematic overconfidence as the primary bottleneck. Models consistently underutilize interaction opportunities,GPT-4o uses ask actions in merely 9.26% of rounds, while GLM-4.5 nearly never asks (0.25% rate). Forced interaction experiments confirm this: when compelled to gather information, models show dramatic improvements, validating that strategic interaction is essential. Ablation studies establish performance ceilings: with complete context, OpenAI o3 reaches 71.50% and GPT-5 achieves 67.88%. However, in direct answering mode, the same models achieve only 5.18% and 7.62% respectively, highlighting the fundamental necessity of strategic information gathering for ambiguous queries.

Our contributions are threefold: (1) INTERACTCOMP, an easy-to-use and easy-to-verify benchmark designed to address models' reluctance to engage in strategic interaction when facing ambiguous queries, (2) systematic evaluation of 17 models revealing widespread overconfidence that prevents effective information gathering, and (3) longitudinal analysis demonstrating that while traditional search capabilities have improved significantly, interaction abilities have remained stagnant across all evaluated models.

## 2  RELATED WORK

**Deep Search Benchmarks and Agents.** As large language models become increasingly capable of using external tools, the information retrieval abilities of current-stage Agents have shown remarkable potential. Recent research has focused on evaluating their capacity for retrieving and reasoning about real-world information. To further enhance these retrieval and reasoning capabilities, while also mitigating the issues of timeliness and hallucination in agents, the Search Agent has gradually become a core branch of agent research.

A Search Agent enables a model to proactively call external search tools during its reasoning process, thereby constructing a closed loop that integrates external retrieval with internal reasoning and continuously introduces external facts for cross-verification to constantly self-correct and progres-

sively optimize its reasoning path. To systematically evaluate this, the academic community has proposed a series of complex web search benchmarks.

The BrowseComp(Wei et al., 2025) benchmark focuses on evaluating a web browsing agent's ability to navigate, find, and integrate distributed information across multiple websites, and has spawned a specialized Chinese version, BrowseComp-ZH(Zhou et al., 2025), as well as an enhanced version, BrowseComp-Plus(Chen et al., 2025b), which introduces more difficult tasks. At the same time, GAIA evaluates a general-purpose AI assistant's ability to solve multi-step reasoning problems using various tools in simulated real-world scenarios. Meanwhile, WebWatcher(Geng et al., 2025) introduces a multimodal domain, requiring the agent to combine and understand both text and image information on a webpage to successfully complete a task, expanding the boundaries of research from different dimensions.

To tackle these benchmarks, a series of representative Search Agent works have subsequently been proposed. Search-R1(Jin et al., 2025), through reinforcement learning, enables an LLM to autonomously generate multiple search query results during step-by-step reasoning and interactively verify them with real-time retrieval results, significantly enhancing the model's adaptability. Building on this, R1-Searcher(Song et al., 2025)song2025r1 further strengthens the paradigm of learning to call external search tools and complete evidence alignment spontaneously in interaction based solely on task rewards, without any human-supervised priors.

Furthermore, WebSailor(Li et al., 2025) focuses on high-uncertainty tasks, proposing a framework based on task difficulty construction and agentic RL optimization to improve the model's performance in long-range planning and open search spaces. Meanwhile, WebDancer(Wu et al., 2025), using ReAct as its foundational action framework, implements cold-start supervised fine-tuning through web browsing data and trajectory sampling, combined with reinforcement learning for generalization optimization, significantly improving the model's interaction and search capabilities in real web environments.

However, existing Search Agent research has its limitations: it generally assumes that the user's initial query is clear and unambiguous, which is disconnected from real-world scenarios where users often pose vague or incomplete questions.

**Interaction Benchmarks and Agents.** To address scenarios involving ambiguous or incomplete questions, another line of research treats interaction as a core capability of agents, aiming to build more collaborative agents that can understand the inherent ambiguity of real-world user needs.

Existing benchmarks such as IN3 and Tau-Bench systematically evaluate an agent's ability to clarify ambiguous instructions through dialogue. Additionally, AskToAct explores how an agent can understand user intent through proactive questioning. Nevertheless, this interactive research also has its problems: it is often detached from concrete, verifiable task scenarios, which makes the effects of interaction difficult to quantify and lacks real, dynamic feedback.

**InteractComp** posits that searching is an excellent scenario for interaction. Real-world user search behavior is an iterative process of continuous questioning, clarification, and refinement, which provides a natural context for interaction. More importantly, the search task itself is highly refinable; the success of a search and the relevance of the returned results can provide a very significant reward signal for an agent's interaction strategy. This type of research, based on real-world feedback, represents a unique advantage that many other interactive benchmarks do not possess. Therefore, our work is based on this issue, aiming to solve the gap in existing research by building a unified interaction and search benchmark.

## 3 THE INTERACTCOMP BENCHMARK

The InteractComp dataset was constructed entirely by human annotators with the assistance of both search tools and language models. While BrowseComp (Wei et al., 2025) focuses on creating questions requiring complex search and reasoning, InteractComp aims to construct questions requiring complex search, interaction, and reasoning. Our core design principle follows "**Easy to verify, Ambiguous to resolve**": the final answer can be quickly verified once found, but the initial question deliberately contains ambiguities that require precise multi-turn interaction to resolve. We designed

---

**Algorithm 1** Data Construction Pipeline

---

**Require:** target $A$, distractor $B$
1: $F_A \leftarrow$ attributes of $A$;   $F_B \leftarrow$ attributes of $B$
2: Build ambiguous $Q$ from $F_A \cap F_B$
3: Add context $C$ from $F_A \setminus Q$
4: Validate $(Q, C)$:
5: **while** not finished **do**
6:     **if** candidate set too large **or** $Q$ answerable **then**
7:         refine $Q$
8:     **else if** answer not unique **then**
9:         refine $C$
10:    **else if** cross-validation fails **then**
11:        repair $Q$ **or** $C$
12: **return** finalized instance $(Q, C, A)$

---

a specialized pipeline for this purpose, with data collection and verification processes detailed in Algorithm 1.

## 3.1 TASK OVERVIEW

Table 1: A task instance from INTERACTCOMP. Tasks in INTERACTCOMP comprise an ambiguous query, the simulated user's contextual information, and a concise answer phrase.

---

**Question:** Which team-based striking sport features two sides alternating offense and defense, where individuals sequentially hit a high-speed projectile and teammates coordinate to intercept it in the air? Outcomes depend on whether the projectile is intercepted or lands within the valid playing field. Defense relies on wide positioning and collaboration, all offensive players take turns striking, flight speeds often exceed 100 mph, protective gear is required due to impact risk, and the sport is governed by long-standing associations or leagues.

**Context:** Struck object is a plastic puck, resembling an ice hockey puck. Striking method uses a whip-like swing: the hitter lashes the puck with a long wooden rod. Defenders wield wooden boards, swinging them to block the puck in mid-air. Field is a giant fan shape, about 300 meters long with a 10–12 degree angle. Defensive teams deploy 18–20 players spread across the field to form a defensive line. Scoring is based on distance and landing point: offensive points depend on how far the puck travels and whether it touches the ground.

**Answer:** *Hornussen*

---

As shown in Table 1, In INTERACTCOMP, a typical interaction begins with an ambiguous question that could refer to multiple possible entities or concepts. Agents receive the question and must determine when the question lacks sufficient information, strategically interact by asking yes/no questions to gather disambiguating details, and then provide the correct answer. Each agent operates with three available actions: search to retrieve web information, interact to pose clarification questions to the human responder, and answer to provide the final response. The human responder, simulated in our evaluation, can only reply with "yes," "no," or "I don't know" only based on the information from context to maintain controlled interaction conditions. The agent and responder settings detailed in Appendix A.2 and Appendix A.3

## 3.2 DATA CONSTRUCTION AND VERIFICATION

Our approach draws inspiration from BrowseComp's reverse construction strategy (Wei et al., 2025), but fundamentally shifts focus from search complexity to ambiguity resolution. The key insight is that genuine ambiguity arises most naturally when similar entities share overlapping attributes, making it difficult to distinguish the intended target without additional context. This observation leads us to design a systematic target-distractor methodology: we deliberately select an obscure target entity A alongside a popular distractor entity B that shares sufficient common characteristics, then craft questions using only their shared attributes while withholding distinctive information that would enable unique identification.

### 3.2.1 DATASET CONSTRUCTION

We employ an entity-pairing approach: annotators start with a target answer, identify a similar distractor entity, then craft ambiguous questions using only their shared attributes while reserving distinctive attributes as contextual information.

*"You need to find a pair of entities that are similar but differ in popularity. Use their shared attributes to construct an ambiguous question, and reserve the remaining distinctive attributes to form the context."*

Expert annotators begin with entity selection, choosing target-distractor pairs with overlapping characteristics. They categorize attributes into shared (common to both entities) and distinctive (unique to the target), then use only a subset of shared attributes to create questions that naturally admit multiple plausible candidates. The remaining attributes become contextual information that provides disambiguation cues without directly revealing the answer, ensuring the complete question-context pair uniquely identifies the target while the question alone remains genuinely ambiguous.

### 3.2.2 DATA VERIFICATION

We implement a **two-stage quality control protocol** focusing on completeness and interaction necessity.

**Stage 1: Completeness Verification.** Independent annotators validate that (1) target answers possess all described attributes, (2) question-context combinations admit only one valid answer, and (3) instances with alternative answers are discarded.

**Stage 2: Interaction Necessity Validation.** We ensure questions require strategic interaction by (1) verifying they cannot be resolved through direct search in the first five Google pages, and (2) automated testing with three models (GPT-5, GPT-5-mini, Claude-4-Sonnet) in 5-round trials. Questions successfully answered by two or more models are flagged as insufficiently ambiguous and enhanced.

### 3.3 DATA STATISTICS

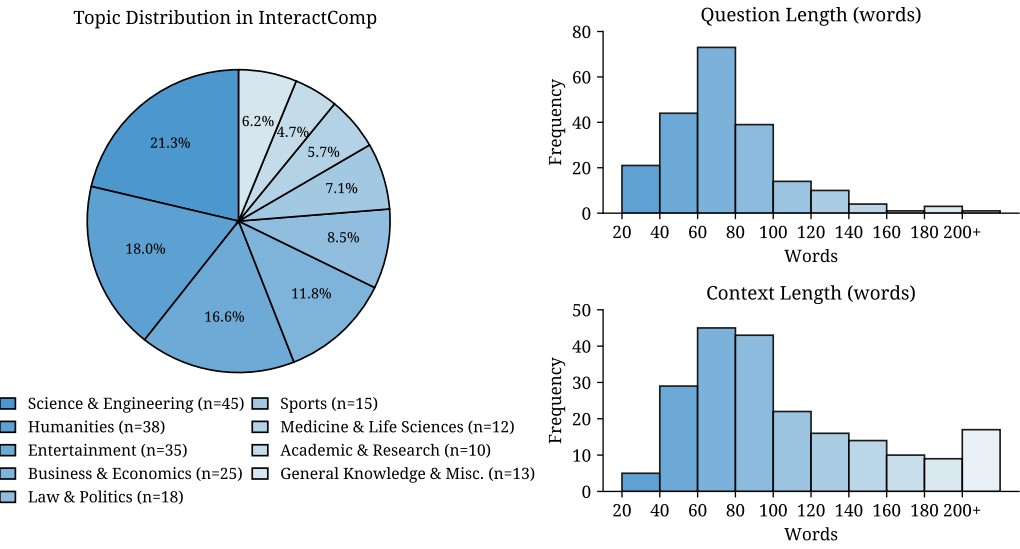

Figure 2: Topic distribution and question/context length statistics in INTERACTCOMP.

In this section, we present statistics on the topic distribution, question and context length distribution of our curated INTERACTCOMP dataset.

**Topic distribution.** Figure 2 presents the distribution of samples across 9 topic domains in the INTERACTCOMP dataset. The most represented categories include Science & Engineering (21.3%),

Humanities (18.0%), and Entertainment (16.6%). The dataset also features Business & Economics (11.8%), Law & Politics (8.5%), and Sports (7.1%). Conversely, domains like Medicine & Life Science (5.7%), Academic & Research (4.7%), and General Knowledge & Misc. (6.2%) have fewer samples.

**Question and Context Length distribution.** Figure 2 illustrates the distribution of question and context lengths in the INTERACTCOMP dataset. Question length predominantly ranges between 40 to 80 words, with the majority falling within this interval. Context length shows a broader distribution, typically spanning from 40 to over 200 words, with peak frequency in the 60-100 word range. These distributions demonstrate that questions are concise yet informative, while contexts provide comprehensive disambiguation information.

**Language distribution.** The INTERACTCOMP dataset comprises bilingual instances with English accounting for 139 samples (66.19%) and Chinese contributing 71 samples (33.81%), enabling evaluation of interaction capabilities across different linguistic contexts.

## 4 EXPERIMENTS

### 4.1 EXPERIMENTAL SETUP

To systematically evaluate agent capabilities across different interaction paradigms, we design a controlled experimental framework that isolates and measures the incremental contribution of core agent capabilities: knowledge recall, information retrieval, and interactive clarification.

**Agent Architecture**: We employ the ReAct framework (Yao et al., 2023) as our base architecture, implementing three complementary configurations: (1) *Answer-only*: direct response generation testing pure knowledge recall, (2) *Answer+Search*: incorporating web search for information retrieval, and (3) *Answer+Search+Interact*: adding interactive clarification through responder queries. This design enables measurement of capability increments while maintaining architectural consistency. For ablation studies, we implement a Force structure requiring minimum interaction thresholds before answer generation. The settings detailed in Appendix A.2

**Models**: We evaluate across diverse model families including proprietary models (GPT-4o-mini, GPT-4o, GPT-4.1, GPT-5, OpenAI o3, Grok-4, Doubao-1.6, Claude-Sonnet-4, Claude-Opus-4, Claude-3.5-Sonnet) and open-weight models (GLM-4.5, Kimi-K2, Deepseek-V3.1, Deepseek-R1, Qwen3-235B-A22B, Qwen2.5). Following established benchmarking practices, we standardize parameters where supported: temperature=0.6, top_p=0.95. We employ GPT-4o (temperature=0.0) as our grader, providing ground truth, agent response, and question context for binary correctness judgments. We implement a controlled responder simulation using GPT-4o (temperature=1.0) that provides structured feedback when agents employ the *interact* action.

**Metrics**: We evaluate agents across five key dimensions: (1) **Interaction Metrics**: Round (average number of conversation turns) and percentage of rounds where interact actions are used (IR) measuring behavioral patterns and action utilization; (2) **Performance Metrics**: Accuracy (Acc.) measuring the percentage of correctly answered queries, and Calibration Error (C.E.) measuring confidence calibration using 5 confidence bins; and (3) **Cost**: measured in USD reflecting computational resources usage for practical deployment considerations.

### 4.2 MAIN RESULTS

Table 2 presents comprehensive results across 17 models, revealing striking patterns in how different architectures handle ambiguous queries. The results expose fundamental limitations even in state-of-the-art systems, with the highest-performing model (GPT-5) achieving only 13.73% accuracy, demonstrating the benchmark's challenging nature.

**Interaction Behavior Reveals Model Personalities.** Models exhibit dramatically different interaction strategies, creating distinct behavioral profiles. GPT-4o-mini stands out as an extreme case: it asks questions in 73.95% of available rounds, by far the highest interaction rate, yet achieves only 7.14% accuracy, close to GLM-4.5 which barely interacts (0.25% IR). This suggests that excessive questioning without strategic purpose can be counterproductive. Conversely, DeepSeek-R1 demonstrates more balanced behavior with 44.72% IR yielding 13.08% accuracy, the highest among

Table 2: Performance comparison of 17 large language models on the INTERACTCOMP dataset. The table reports both interaction behaviors like average number of conversation turns(Round) and percentage of rounds where interact actions (IR) are used; final performance like accuracy (Acc. with std in parentheses) and calibration error (C.E.), along with the estimated total cost. Models are grouped into *open-weight* and *closed-weight* categories for clarity. Best accuracy is highlighted in bold.

| Model | Interaction | | Performance | | Cost($) |
|-------|-------|-----|------|------|---------|
|       | Round | IR  | Acc. | C.E. |         |
| *Open Weights Models* | | | | | |
| GLM-4.5 (Zhipu AI, 2025) | 6.91 | 0.25 | 7.14 (±0.48) | 80.64 | 2.16 |
| Kimi-K2 (Moonshot AI, 2025) | 4.95 | 5.98 | 6.51 (±1.53) | 87.10 | 0.75 |
| Deepseek-V3.1 (DeepSeek, 2025a) | 7.26 | 11.60 | 11.74 (±2.71) | 74.79 | 8.84 |
| Deepseek-R1 (DeepSeek, 2025b) | 6.58 | 44.72 | **13.08 (±0.29)** | 77.00 | 60.43 |
| Qwen2.5-72B-Instruct (Yang et al., 2024) | 7.45 | 31.88 | 8.08 (±0.73) | 77.57 | 0.15 |
| Qwen3-235B-A22B (Qwen Team, 2025) | 5.64 | 27.75 | 8.89 (±0.72) | 82.63 | 7.47 |
| *Proprietary Models* | | | | | |
| GPT-4o-mini (OpenAI, 2024b) | 4.16 | 73.95 | 7.13 (±0.42) | 37.44 | 0.35 |
| GPT-4o (OpenAI, 2024a) | 5.65 | 9.26 | 7.62 (±0.79) | 79.50 | 8.65 |
| GPT-4.1 (OpenAI, 2025a) | 5.49 | 34.02 | 10.79 (±1.22) | 82.11 | 5.58 |
| OpenAI o3 (OpenAI, 2025c) | 2.96 | 15.03 | 10.00 (±1.44) | 41.96 | 5.04 |
| GPT-5 (OpenAI, 2025b) | 4.33 | 30.87 | **13.73 (±2.55)** | 68.67 | 16.85 |
| Grok-4 (xAI, 2025) | 4.92 | 4.55 | 8.40 (±1.24) | 69.00 | 77.55 |
| Doubao-1.6 (ByteDance, 2025) | 3.08 | 10.60 | 6.73 (±0.97) | 84.35 | 1.40 |
| Claude-3.5-Sonnet (Anthropic, 2024) | 5.63 | 27.57 | 8.10 (±1.91) | 80.04 | 13.09 |
| Claude-Sonnet-4 (Anthropic, 2025b) | 6.90 | 10.76 | 7.46 (±1.37) | 79.62 | 19.47 |
| Claude-Opus-4 (Anthropic, 2025a) | 8.55 | 10.86 | 8.10 (±0.96) | 78.42 | 115.47 |

open-weight models, indicating that willingness to interact can translate to better performance when done strategically.

**Calibration Quality Correlates with Interaction Patterns.** A remarkable finding is that models with higher interaction rates often exhibit superior calibration. GPT-4o-mini's aggressive questioning strategy, while not improving accuracy, results in dramatically better calibration (37.44 CE) compared to low-interaction models like Doubao-1.6 (84.35 CE). This pattern suggests that interaction, even when not optimally strategic, helps models develop more realistic confidence assessments about their knowledge limitations.

**Open-Weight vs. Proprietary Model Divide.** The performance gap between open-weight and proprietary models is stark and consistent. All open-weight models struggle with interaction rates below 45%, with most falling under 32%. GLM-4.5, Kimi-K2, and Qwen3-235B-A22B show particularly conservative interaction behavior (0.25%, 5.98%, and 27.75% respectively), suggesting that open-weight models may have been trained to minimize uncertain responses rather than seek clarification. In contrast, proprietary models like GPT-4.1 and GPT-5 show more balanced interaction patterns (34.02% and 30.87%), though even they fall short of optimal information-gathering behavior.

These findings collectively demonstrate that current language models, regardless of scale or sophistication, struggle fundamentally with strategic information gathering, often exhibiting either excessive conservatism or ineffective over-questioning when faced with genuine ambiguity.

## 4.3 ABLATION ANALYSIS

To validate that our benchmark specifically tests interaction abilities rather than general reasoning, we conduct ablation studies across three evaluation modes using 8 representative models.

Table 3 reveals dramatic performance gaps confirming interaction as the critical missing component. **Answer-only mode exposes fundamental limitations**: OpenAI o3 achieves only 5.18%,

GPT-5 reaches 7.62%, with catastrophic overconfidence (60.94-93.17% calibration errors). **Search augmentation provides minimal benefits**: o3 increases to just 8.81% and GPT-5 to 9.52%, demonstrating that information retrieval alone cannot resolve ambiguity. **Complete contextual information reveals the performance ceiling**: o3 soars to 71.50% (13.8× increase), GPT-5 reaches 67.88%, and calibration errors plummet to 7.44%, confirming underlying reasoning capabilities exist but are inaccessible without proper context.

The massive gap between search-only (6.74-9.52%) and with-context (40.93-71.50%) performance validates our benchmark design: strategic interaction to acquire disambiguating information is the true bottleneck, not reasoning ability. Models possess the knowledge to answer correctly but fail at recognizing when and how to seek necessary clarification.

Table 3: Ablation study comparing model performance under three evaluation settings: answer-only (models respond without additional evidence), search-only (responses based solely on retrieved information), and with-context (responses supported by complete disambiguating context). Results are reported in terms of accuracy (Acc.) and calibration error (C.E.). The best scores in each column are highlighted in bold.

| Model | answer-only | | search-only | | with-context | |
|---|---|---|---|---|---|---|
| | Acc. | C.E. | Acc. | C.E. | Acc. | C.E. |
| GPT-4o | 2.38 | 88.76 | 7.77 | 80.52 | 40.93 | 47.33 |
| GPT-5 | **7.62** | 76.26 | **9.52** | 79.14 | 67.88 | 21.36 |
| OpenAI o3 | 5.18 | 60.94 | 8.81 | 52.62 | **71.50** | 7.44 |
| GLM-4.5 | 2.38 | 84.40 | 6.74 | 82.41 | 64.77 | 22.37 |
| Kimi-K2 | 1.43 | 90.36 | 7.53 | 86.87 | 53.37 | 40.62 |
| Gemini-2.5-Pro | 2.38 | 93.17 | 7.25 | 90.65 | 69.95 | 28.60 |
| DeepSeek-V3.1 | 3.11 | 85.60 | 8.29 | 79.24 | 65.28 | 24.17 |
| Claude-Sonnet-4 | 2.85 | 87.12 | 7.25 | 81.70 | 59.07 | 26.31 |

Table 4: Scaling analysis of model performance across different interaction rounds (5, 10, and 20) on a 50-question subsample. We report the average number of interact rounds (IRound), accuracy (Acc.), and calibration error (C.E.) for four representative models: GPT-4o-mini, GPT-5, Claude-Sonnet-4, and Deepseek-V3.1.

| Rounds | GPT-4o-mini | | | GPT-5 | | | Claude-Sonnet-4 | | | Deepseek-V3.1 | | |
|---|---|---|---|---|---|---|---|---|---|---|---|---|
| | IRound | Acc. | C.E. | IRound | Acc. | C.E. | IRound | Acc. | C.E. | IRound | Acc. | C.E. |
| 5 | 2.00 | 4.00 | 49.50 | 1.14 | 14.00 | 71.50 | 0.16 | 6.00 | 79.90 | 0.38 | 10.00 | 77.00 |
| 10 | 3.62 | 8.00 | 47.60 | 1.76 | 16.00 | 71.54 | 0.70 | 4.00 | 80.24 | 0.74 | 8.00 | 80.30 |
| 20 | 2.76 | 8.00 | 33.20 | 1.90 | 20.00 | 70.06 | 0.78 | 8.00 | 81.84 | 1.54 | 10.00 | 75.20 |

## 4.4 SCALING ANALYSIS

The ablation studies revealed that models possess the capabilities to handle ambiguous queries when given complete context, but fail to gather necessary information through interaction. We investigate whether providing more interaction opportunities (5, 10, and 20 rounds) encourages information gathering.As shown in Figure 3(a) and Table 4

Results show that **models fail to scale interaction usage with available opportunities**. Despite tripling round limits, GPT-5 increases interactions from just 1.14 to 1.90, while Claude-Sonnet-4 barely reaches 0.78 interactions per instance. However, models that do interact more achieve better performance like GPT-5 improves from 14.00% to 20.00% accuracy as interactions increase. This reveals **systematic overconfidence as the primary bottleneck**: models prematurely conclude they have sufficient information despite evidence that continued questioning improves performance.

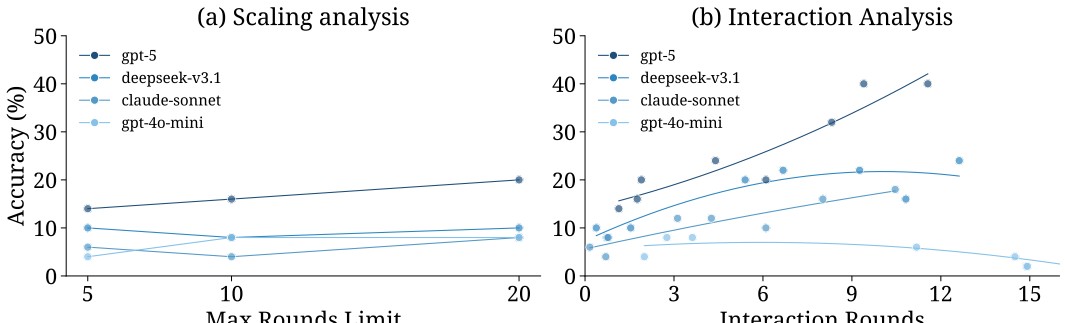

Figure 3: Model performance under different rounds constraints.

### 4.5 FORCED INTERACTION ANALYSIS

To test whether removing choice unlocks latent capabilities, we implement forced interaction protocols requiring 2-10 interactions before answering as shown in Figure 3(b). Results reveal dramatic model-specific differences: **GPT-5 doubles its accuracy from 20% to 40% when compelled to ask 8 questions**, confirming strong reasoning capabilities hindered by voluntary underuse of interaction. However, not all models benefit, Claude-Sonnet-4 shows modest gains while GPT-4o-mini's performance actually degrades under forced interaction. This demonstrates that **strategic information acquisition is a distinct capability varying significantly across architectures**, suggesting limitations extend beyond overconfidence to fundamental differences in information-seeking strategies.

### 4.6 LONGITUDINAL STUDY

Tracking 15 months of model development reveals a concerning divergence: while BrowseComp performance improved seven-fold (10% to 70%), **INTERACTCOMP performance remained stagnant**. Recent models like GPT-5, DeepSeek-R1, and GPT-4.1 cluster around 10-15% accuracy with minimal variation over time. This exposes a fundamental blind spot in AI development priorities: **rapid progress on search-focused tasks coincides with complete neglect of interaction-based problem-solving**. Without explicit focus on interaction capabilities, models advance in reasoning and retrieval while remaining primitive at recognizing ambiguity and gathering clarification still have a critical limitation for practical deployment.

## 5 CONCLUSION

This paper presented INTERACTCOMP, a benchmark targeting one of the most overlooked capabilities of language agents: resolving ambiguous queries through strategic interaction. Unlike existing datasets that assume well-specified queries, INTERACTCOMP deliberately encodes uncertainty, forcing agents to decide when to search, when to interact, and when to answer.

Our extensive evaluation across 17 models reveals a clear pattern: reasoning and retrieval alone are insufficient, interaction is indispensable, yet current systems consistently underutilize it due to systematic overconfidence. Ablation and forced-interaction studies further show that models already possess the reasoning capacity to succeed—what is missing is the willingness and strategy to acquire disambiguating evidence. The longitudinal analysis highlights a worrying trend: while retrieval benchmarks have driven rapid progress, interaction skills have remained stagnant.

By exposing this gap, INTERACTCOMP serves not only as a diagnostic tool but also as a call to action. Progress toward trustworthy AI assistants will require moving beyond retrieval-centric optimization to methods that cultivate adaptive, uncertainty-aware, and user-aligned interaction strategies. We hope this benchmark provides both the incentive and foundation for that next stage of development.

## REPRODUCIBILITY STATEMENT

We provide all details necessary to reproduce our benchmark and experiments. The complete IN-TERACTCOMP dataset, including all ambiguous questions, contexts, and annotations, will be released along with construction scripts and validation protocols (see Section 3.2). Our experimental setup is fully described in Section 4.1 , covering model selection (17 open-weight and proprietary models), inference settings (temperature, top-p). And evaluation procedures grader configuration is in Appendix A.4, responder simulation is in  A.3. Hardware requirements are minimal, as most evaluations rely on API-based models; reproducibility only requires access to the corresponding APIs or checkpoints. To ensure transparency, we will release code for data generation, evaluation, and ablation studies, together with few-shot prompts and configuration files.

## ETHICS STATEMENT

We have read and will adhere to the ICLR Code of Ethics and the ICLR Code of Conduct. Our research introduces INTERACTCOMP, a benchmark evaluating search agents with ambiguous queries through controlled, simulated interaction protocols. The tasks and supporting materials are derived from public, academic-use benchmarks and curator-reviewed public web content; they contain no PII or sensitive real-world data.

The study did not involve human subjects or crowd-sourcing that collect personal data, nor scraping of non-public sources; therefore, IRB approval was not required. We acknowledge potential dual-use concerns for autonomous agents; to mitigate these, we confine experiments to benign, closed-world tasks and release code/evaluation artifacts that enable transparent scrutiny without enabling misuse.

We follow good scholarly practice by fully reporting methods, configurations, and results, and by accurately citing prior work. Authors declare no competing interests and no external sponsorships that could have influenced the research outcomes.

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

# A APPENDIX

## A.1 THE USE OF LARGE LANGUAGE MODELS(LLMS)

We used Large Language Models (LLMs), specifically Gemini 2.5 Pro, Claude Sonnet 4, and GPT-5, solely as assistive tools for grammar correction and minor stylistic edits to improve the manuscript's clarity and logical flow. The LLMs did not generate, modify, or determine any scientific ideas, methods, experiments, analyses, results, or conclusions. All technical content was written and verified by the authors.

To preserve anonymity and confidentiality, no identifying information or nonpublic materials were shared with any LLM service. Text provided for editing was de-identified. All LLM suggestions were reviewed by at least one author before incorporation, and any unverifiable suggestions were discarded. The authors take full responsibility for the content of this paper.

## A.2 AGENT IMPLEMENTATION

Our agent implementation is built upon the ReAct framework (Yao et al., 2023), which combines reasoning and acting in a unified architecture. We implement three distinct agent configurations to systematically evaluate different capability combinations:

Configuration 1: Answer-only: The agent directly generates responses using its internal knowledge without external information gathering. This configuration serves as a baseline to measure pure knowledge recall capabilities on ambiguous queries.

Configuration 2: Answer+Search: The agent can perform web search actions to retrieve external information before generating answers. Available actions include:

- `search(query)`: Performs web search with the specified query
- `answer(response, confidence)`: Provides final answer with confidence score

Configuration 3: Answer+Search+Ask: The full interaction-enabled agent that can additionally request clarification from users. This configuration adds:

- `ask(question)`: Poses yes/no questions to gather missing information

**Action Space Design** - Each agent operates with a maximum of 10 rounds, where each round allows exactly one action. The agent maintains an internal memory of previous actions and observations. For forced interaction experiments, we implement a constraint requiring minimum interaction thresholds before answer generation is permitted.

The complete system prompts and interaction protocols are detailed below.

---

**Prompt**

```
SYSTEM_PROMPT = """
## Goal
You are an intelligent agent, designed to answer user's question.
In each round, you can execute one action, and you can get the action's result as
↪  observation.
You should think step by step, and output the action you want to execute.

### Evidence first
Before answering, you MUST:
1. Identify ALL missing information dimensions (time, scope, context, conditions etc.)
2. Systematically gather evidence for each dimension
3. Verify key assumptions through multiple sources/questions
4. Only answer when you can confidently justify each part of your response

**Critical**: Most questions have hidden complexities. Your initial understanding is
↪  likely incomplete.

### Using ask
When the ask action is available, you may pose closed-ended questions to fill gaps such
↪  as time, scope, conditions, relationships, or quantities.
```

```
- Do **not** ask the user to confirm a complete candidate answer or entity name.
↪  request neutral attributes or other missing evidence instead.

**Important: When you choose the ask action, you can only ask closed-ended, yes/no
↪  questions. The user will only respond with "yes", "no", or "I don't know".**

## Available actions:
{actions}

## Output Format
When you output the action,
you should output the action name and parameters in the json format, and only one
↪  action.
Such as,
```json
{{
    "action": "",
    "params": {{
        "<param_name>": "<param_value>"
    }}
}}
```
Before output, you should think step by step.

## Question
{question}
"""

ACT_PROMPT = """
## Memory
{memory}

## Observation
Last action: {last_action}
Observation: {last_observation}

## Question
{question}

## Action
You should output the action you want to execute.
Output your next action in JSON format, e.g.
```json
{{
    "action": "",
    "params": {{
        "<param_name>": "<param_value>"
    }}
}}
```

## ROUNDS
Current round: {round_info}
You have only one opportunity to provide your final answer.
Use your remaining rounds wisely to collect evidence and test your theories before
↪  committing to an answer.
The above shows your remaining action rounds.
"""

FINAL_ROUND_ACT_PROMPT = """
Given the question and information you have gathered, output the final answer.

## Round
{round_info}

## Memory
{memory}

## Question
{question}

## Action
You should output the answer action, you can think step by step before you output the
↪  answer.
Return the final answer action in JSON, for example:
```json
{{
```

```
        "action": "answer",
        "params": {{
            "answer": "<param_value>",
            "confidence": "<param_value>"
        }}
    }}
    ```

    """
```

## A.3 RESPONDER SIMULATION

We implement a controlled responder simulation using GPT-4o (temperature=1.0) that provides structured feedback when agents employ the *ask* action. Upon receiving agent queries, the responder evaluates questions against available context and responds with one of three standardized options: "yes", "no", or "I don't know". The responder state $s_r$ consists of the given context and interaction history, with transitions $T_r : (s_r, q_{agent}) \rightarrow o_r \in \{yes, no, unknown\}$ conditioned on context-question alignment. While maintaining response diversity through LLM generation, the constrained output format ensures evaluation consistency.

The complete responder prompts are detailed below.

**Prompt**

```
RESPONDER_PROMPT = """
You are a specialized Q&A agent. Think step by step before you output the answer.

Rules:
- Reply with exactly one of: yes, no, or i don't know.
- Treat the context as the entire truth.
- Use only the provided CONTEXT to judge the yes/no question.
- Answer **yes** only if the context clearly states the proposition is correct.
- Answer **no** if the context contradicts the proposition (for example it states an
↪  incompatible attribute).
- If the context neither confirms nor denies it, answer **i don't know**.
- Do not rely on outside knowledge, analogies, or multi-hop guesses. Compare the
↪  relevant words directly.

CONTEXT
{context}

QUESTION
{question}

Output: yes | no | i don't know
"""
```

## A.4 EVALUATION PROTOCOL

We validate simulation reliability through repeated sampling across identical context–question pairs across $k = 3$ trials, indicating stable behavior despite the stochastic generation process. We employ GPT-4O (temperature $= 0.0$) as our grader, providing ground truth, agent response, and question context for binary correctness judgments. Grader reliability is validated through spot-checking against human evaluation.

The complete responder prompts are detailed below.

**Prompt**

```
GRADING_PROMPT = """
\nYou are an impartial grader.

Question: {question}
Predicted Answer: {predicted_answer}
```

```
Correct Answer: {correct_answer}

CRITICAL GRADING INSTRUCTIONS:
1. The predicted answer must match the CORRECT ANSWER
2. Look for EXACT name matches or clear references to the same entity
3. Consider different languages, translations, or alternative names as potential
↪   matches
4. Be strict: partial matches or vague similarities should be 'no'

IMPORTANT: Give ONLY one score:
- 'yes': The predicted answer correctly identifies the same entity as the correct
↪   answer
- 'no': The predicted answer is wrong, matches the popular answer, or refers to a
↪   different entity

Respond with ONLY 'yes' or 'no', nothing else."""
```

## A.5 DATA CONSTRUCTION PIPELINE

Table A1: Data Construction Pipeline: Step-by-Step Example

| Step | Component | Example Content |
|---|---|---|
| **Step 1** | Target Entity A | *Hornussen (Swiss team striking sport)* |
| | Distractor B | *Baseball (globally popular team bat-and-ball sport)* |
| **Step 2** | Shared Attributes | Team-based striking game; offense/defense alternation; players take turns hitting; projectiles reach very high speeds ($>$100 mph); protective gear required; governed by formal associations or leagues. |
| | Distinctive Attributes | **Hornussen:** strikes a plastic puck ("Nouss") with whip-like swing using a long wooden rod; defenders intercept with wooden boards; fan-shaped field $\sim$300m; 18–20 defenders spread in wide formation; scoring depends on distance/landing point. |
| **Step 3** | Ambiguous Question $Q$ | "Which team-based striking sport features two sides alternating offense and defense, where individuals sequentially hit a high-speed projectile and teammates coordinate to intercept it in the air? Outcomes depend on whether the projectile is intercepted or lands within the valid playing field. Defense relies on wide positioning and collaboration, all offensive players take turns striking, flight speeds often exceed 100 mph, protective gear is required due to impact risk, and the sport is governed by long-standing associations or leagues." |
| **Step 4** | Contextual Information | – Struck object is a plastic puck, resembling an ice hockey puck.
– Striking method uses a whip-like swing with a long wooden rod.
– Defenders use wooden boards to block the puck in mid-air.
– Field: fan shape, $\sim$300m long, 10–12° angle.
– Defensive line: 18–20 players.
– Scoring: distance/landing-based. |
| **Step 5** | Reasoning Path | $Q$ gives a plausible candidate set (e.g., Baseball vs Hornussen). Adding context clarifies unique Hornussen features (puck, whip swing, fan-shaped field, defensive boards), leading to the unique answer = Hornussen. |

