# OpenReview forum: "InteractComp: Evaluating Search Agents With Ambiguous Queries"
_ICLR.cc/2026/Conference — ICLR 2026 Conference Withdrawn Submission_

### Official Review · Reviewer_dMTZ · 2025-10-28

**Soundness:** 2
**Presentation:** 2
**Contribution:** 3
**Rating:** 4
**Confidence:** 4

**Summary:**

1. This paper introduces InteractComp, a benchmark designed to evaluate whether search agents can recognize and resolve ambiguous user queries through strategic interaction.

2. The InteractComp comprises 210 expert-curated questions, which are annotated by humans using an entity-pairing approach and a two-stage quality control protocol.

3. Experiments on 17 models show that even the strongest models perform poorly mainly due to overconfidence and underuse of interaction.

**Strengths:**

1. This paper demonstrates a good motivation, highlighting that human search behavior is typically iterative, beginning with ambiguous queries and progressively refining them through interaction. This perspective makes the evaluation setting more closely aligned with real-world scenarios.

2. The data construction is based on an insightful idea: ambiguity arises when similar entities share overlapping attributes. The benchmark also considers domain generalization and includes data quality verification, which strengthens its credibility.

3. The evaluation covers a wide range of models, including both proprietary and open-source LLMs, under different configurations and model sizes, providing a comprehensive comparison and analysis. The final conclusions attribute model failures to overconfidence and reluctance to interact, which is well-supported and reasonable.

**Weaknesses:**

1. This paper can essentially be understood as targeting a multi-turn collaborative/conversational search scenario. Although InteractComp is verifiable, its query format (as exemplified in Table 1) does not show a clear distinction from complex search benchmarks like BrowseComp. The core design of BrowseComp involves multiple constraints, many of which are themselves ambiguous. Evidently, both benchmarks inherently possess interactive characteristics.

2. Although I find the heuristic approach for data construction quite insightful, the decision to restrict simulated user interactions to yes/no clarifications only seems rather narrow, especially given the goal of modeling realistic user-agent interaction.

3. The human annotation procedure lacks sufficient detail, making it unclear how well the constructed data aligns with real-world distributions. For example, the principles guiding human experts in selecting entities and attributes remain unspecified.

**Questions:**

1. When a query contains too many constraints, it tends to become excessively long, which introduces a gap from real user scenarios. In InteractComp, is there an emphasis on shorter queries? For example, have you considered questions that do not include all constraints at the first user query but instead require the agent to actively identify ambiguity and ask for clarification?

2. In the data construction process, which parts involve human participation and which rely on LLM-as-Judge? This distinction is important and should be explicitly clarified in the main text.

3. How do you ensure that the ambiguity in each question reflects real semantic ambiguity rather than artificial lexical ambiguity introduced by human phrasing?

4. According to Figure 1(a) in the paper, has the latest model already achieved ~70% accuracy on BrowseComp?

---

### Official Review · Reviewer_fvBE · 2025-10-30

**Soundness:** 3
**Presentation:** 3
**Contribution:** 3
**Rating:** 6
**Confidence:** 4

**Summary:**

Paper introduces a benchmark for testing search and dialogue agents on their ability to recognize and resolve *ambigious* queries. Each benchmark data instance contains (1) an ambiguous question, (2) a context string that contains distinctive attributes that are only revealed to the search agent under simulated interaction via another agent if the search agent asks for it, (3) and a single umambigious correct answer. 210 tasks total, bilingual, across 9 domains.

The dataset is deliberately constructed to be ambiguous through annotator instruction. Across 17 models tested, accuracy remains very low even for SOTA systems.

The paper represents a solid contribution towards more interactive (and realistic) search benchmarks.

**Strengths:**

Addresses a clear gap in current search benchmarks- interactivity. The paper makes a compelling argument that real-life search involves this iterative refinement process, and the paper makes a step in that direction.

**Weaknesses:**

Ecological validity of benchmark construction. Annotators are deliberately asked to construct queries while starting from a target answer. It remains an open question if the scoring well on the benchmark would represent meaningful and demonstrative improvement in real life search tasks. The synthetic interaction channel by forcing a yes/no response from the responder who has the context, is a strange and not fully justified design decision. A more realistic scenario could have the responder respond in free-text (i.e. a search revealed ambiguity, which then revealed a related memory of the object, such that the responder agent could be representative of memory), though controllability would remain harder in that instance.

The paper also misses out on some heavily relevant related work, notably in the "tip of the tongue known-item search" domain. i.e. BLUR (CH-Wang et al, ACL 2025), Tip of the Tongue Known-Item Retrieval: A Case Study in Movie Identification, Arguello et al 2021, etc.

**Questions:**

N/A

---

### Official Review · Reviewer_rgYF · 2025-10-31

**Soundness:** 2
**Presentation:** 2
**Contribution:** 2
**Rating:** 2
**Confidence:** 3

**Summary:**

The paper identifies a gap between the settings of the current search benchmarks and real-world interactions. Current search benchmarks focus on fully-specified unambiguous queries and evaluate the search and reasoning capabilities of search agents, while real-world human queries are often ambiguous and require multi-turn interactions for disambiguation. To this end, they propose a new benchmark, consisting of 210 expert-written ambiguous queries, that cannot be answered without asking for clarifying questions. The authors report that the current state-of-the-art models achieve less than 14% accuracy on the benchmark, with overconfidence and lack of strategic interaction being the main causes of failures.

**Strengths:**

- The paper identifies a gap between current search benchmarks and real-world use cases of search systems.
- To address the above-mentioned gap, the authors construct a new expert-written benchmark that specifically tests the interaction capabilities of the models.
- The developed benchmark is not saturated (~14%) and can be used for evaluating both search and interaction capabilities of the models.
- The authors evaluate both proprietary and open-weight models and identify varying behaviors across different models.

**Weaknesses:**

### Major

**Surface-level analyses**

The analyses of the models is limited to the raw numbers and comparisons across models and ablations. To support the claims and provide explanations of the observed behaviors, a deeper analysis is required. For example: why does increasing the number of interaction turns have different effects on different models?

Additionally, providing qualitative examples can provide intuition and help the researchers better understand the failure modes on the proposed benchmark. Going further, developing a taxonomy of failures can further strengthen the paper, as isolated claims based on the raw numbers are hard to synthesize into generalizable findings.

**Search vs Reasoning/QA vs Interaction**

With the current ablations, it is not possible to separate the effect of the following factors that can potentially impact the final system performance: (1) search module (i.e., search engine), (2) reasoning, QA, and tool-calling capabilities, and (3) interaction capabilities.

What are the proportion of the cases where the system failed due to each of the three factors above? Studying additional ablations in Table 3 can provide more transparecy into the failure modes.

***

### Minor

**Literature review**

- The paper misses literature on real-world human-AI interaction datasets (Wildchat, Chatbot Arena). Specifically, as the paper focuses on real-world ambiguous under-specified queries to search-augmented models, Search Arena, a recent dataset of human interactions with search-augmented LLMs, is highly relevant. Analyzing the ambiguity of natural user-written prompts and using them as part of the benchmark can improve the realism and utility of the benchmark.

- Additionally, the paper does not have citations for the works mentioned in the second part of the Section 2 (Lines 140-144).

**Implementation details**

How is search implemented? What search engine is used?

**Presentation and missing details**

- Algorithm 1 is not properly grounded and explained in text.

- The paper misses explanation of the calibration error (C.E.). There is a brief mention of C.E. in Appendix; however, as it is references multiple times in analysis, it would be helpful to include an explanation in the main body of the paper.

**Questions:**

- Does the `with-context` baseline (Table 3) have access to search? Why are the numbers low even with access to the hidden context? Is it because the failure of the search engine?

- How is search implemented? What search engine is used?

- What are qualitative examples of the failures backing up the claims in the paper (e.g., *"strategic information acquisition is a distinct capability varying significantly across architectures"*)?

---

### Official Review · Reviewer_QM9j · 2025-11-01

**Soundness:** 2
**Presentation:** 2
**Contribution:** 2
**Rating:** 2
**Confidence:** 4

**Summary:**

This work proposes a benchmark for evaluating agents’ abilities to navigate tasks with ambiguities or ambiguous queries. This is an important use case scenario. While the field is working towards fully autonomous agents, it’s also important to think about potential issues of premature commitment, where agents act confidently despite uncertainty. The data is built from 210 bilingual expert-written question-context pairs across nine domains, where each example comes with a yes/no question to distinguish between a correct target and a possible distractor. The authors evaluate 17 models and find that accuracy drops significantly under ambiguity, but improves a lot when agents are forced to ask questions.

**Strengths:**

- The proposed benchmark targets an important scenario: in real life, user queries are often ambiguous or underspecified. To effectively address user queries, models need to know when and how to interact with users to resolve these ambiguities.
- The answers are short and easily verifiable, which reduces subjectivity in evaluation.
- Experiments reveal clear and useful empirical insight that overconfidence and lack of uncertainty awareness is a core weakness.

**Weaknesses:**

- Based on the example in Table 1, the query appears to be something a model could likely resolve using its internal world knowledge rather than through web search. During data construction, there was no verification that these queries require external search to be answered correctly. If that’s the case, the rationale for putting the task in an agentic, search-based setup becomes unclear, especially since prior work has already shown that language models struggle to ask clarification questions.
- There is no human performance baseline that measures: when human users are asked to answer these queries, how often will they answer directly without asking clarification questions? This comparison would help contextualize model behavior and clarify whether current systems truly underperform relative to humans.
- There is no information about how many experts were involved in data creation, which platform they were recruited through, or what their credentials or backgrounds are.
- Important statistics on the dataset are missing: how many items are there per domain? What about language distribution by domain?

**Questions:**

- It would be helpful for the authors to clarify how many examples in the dataset genuinely requires web search to resolve. An analysis quantifying how many items truly depend on external information would strengthen the paper’s positioning.

---

### Note · Authors · 2025-11-28

**Comment:**

We sincerely thank the reviewers for their valuable time and constructive feedback, and we appreciate the ICLR organizing committee for their professional management of the review process.

**Withdrawal Confirmation:**

I have read and agree with the venue's withdrawal policy on behalf of myself and my co-authors.